# Generalisation of structural knowledge in the hippocampal-entorhinal system

**James C.R. Whittington***
University of Oxford, UK
james.whittington@magd.ox.ac.uk

**Timothy H. Muller***
University of Oxford, UK
timothymuller127@gmail.com

**Shirley Mark**
University College London, UK
s.mark@ucl.ac.uk

**Caswell Barry**
University College London, UK
caswell.barry@ucl.ac.uk

**Timothy E.J. Behrens**
University of Oxford, UK
behrens@fmrib.ox.ac.uk

## Abstract

A central problem to understanding intelligence is the concept of generalisation. This allows previously learnt structure to be exploited to solve tasks in novel situations differing in their particularities. We take inspiration from neuroscience, specifically the hippocampal-entorhinal system known to be important for generalisation. We propose that to generalise structural knowledge, the representations of the structure of the world, i.e. how entities in the world relate to each other, need to be separated from representations of the entities themselves. We show, under these principles, artificial neural networks embedded with hierarchy and fast Hebbian memory, can learn the statistics of memories and generalise structural knowledge. Spatial neuronal representations mirroring those found in the brain emerge, suggesting spatial cognition is an instance of more general organising principles. We further unify many entorhinal cell types as basis functions for constructing transition graphs, and show these representations effectively utilise memories. We experimentally support model assumptions, showing a preserved relationship between entorhinal grid and hippocampal place cells across environments.

## 1 Introduction

Animals have a remarkable ability to flexibly take knowledge from one domain and generalise it to another. This is not yet the case for machines. The advantages of generalising knowledge are clear - it allows one to make quick inferences in new situations, without having to always learn afresh. Generalisation of statistical structure (the relationships between objects in the world) imbues an agent with the ability to fit things/concepts together that share the same statistical structure, but differ in the particularities, e.g. when one hears a new story, they can fit it in with what they already know about stories in general, such as there is a beginning, middle and end - when the funny story appears while listening to the the news, it can be inferred that the programme is about to end.

Generalisation is a topic of much interest. Advances in machine learning and artificial intelligence (AI) have been impressive [23, 28], however there is scepticism over whether 'true' underlying structure is being learned. We propose that in order to learn and generalise structural knowledge, this structure must be represented explicitly, i.e. separated from the representations of sensory objects in

the world. In worlds that share the same structure but differ in sensory objects, explicitly represented structure can be combined with sensory information in a conjunctive code unique to each environment. Thus sensory observations are fit with prior learned structural knowledge, leading to generalisation.

In order to understand how we may construct such a system, we take inspiration from neuroscience. The hippocampus is known to be important for generalisation, memory, problems of causality, inferential reasoning, transitive reasoning, conceptual knowledge representation, one-shot imagination, and navigation [13, 7, 17, 25]. We propose the statistics of memories in hippocampus are extracted by cortex [27], and future hippocampal representations/memories are constrained to be consistent with the learned structural knowledge. We find this an interesting system to model using artificial neural networks (ANNs), as it may offer insights into generalisation for machines, further our understanding of the biological system itself, and continue to link neuroscience and AI research [18, 38].

## 1.1 Background

In spatial navigation there is a good understanding of neuronal representations in both hippocampus (e.g. place, landmark cells) and medial entorhinal cortex (e.g. grid, border, object vector cells). Thus when modelling this system, we start with problems akin to navigation so we can both leverage and compare our results to these known representations (noting our approach is more general). Place [29] and grid cells [16] have had a radical impact in neuroscience, leading to the 2014 Nobel Prize in Physiology and Medicine. Place and grid cells are similar in that they have a stable firing pattern for specific regions of space. Place cells tend to only fire in a single (or couple) location in a given environment, whereas grids cells fire in a regular lattice pattern tiling the space. These cells cemented the idea of a 'cognitive map', where an animal holds an internal representation of the space it navigates [36]. Traditionally these cells were believed to be spatial only. It has since emerged that place and grid cells code for both spatial and entirely non-spatial dimensions such as sound frequency [2], and furthermore grid-like codes for two dimensional (2D) non-spatial coordinate systems exist [9]. It therefore seems that place and grid codes may provide a general way of representing information. Other entorhinal cell types (border [31], object vector cells [19]) appear to have disparate roles in coding space. Here we unify them, along with grid cells, as basis functions for transition statistics.

Grid cells may offer a generalisable structural code. Indeed grid cell representations are similar in environments that share structure ([14], section 5). Recent results suggest such codes summarise statistics of 2D space, either via a PCA of hippocampal place cells [12] or as eigenvectors of the successor representation [32]. These summary statistics represent rules of 2D-ness (not just 'spatial' space), e.g. if A is close to B, and B is close to C, we can infer A and C are close. Place cells may offer a conjunctive representation. Their activity is modulated by the sensory environment as well as location [39, 22]. Additionally the place cell code is different for two structurally identical environments - this is called remapping [6, 26]. Remapping is traditionally thought to be random. However, we propose place cells form a conjunctive representation between structural (grid cells) and sensory input, and therefore remap to non-random locations consistent with this conjunction.

## 1.2 Contributions

We implement our proposal in an ANN tasked with predicting sensory observations when walking on 2D graph worlds, where each vertex is associated with a sensory experience. To make accurate predictions, the agent should learn the underlying hidden structure of the graphs. We separate structure from sensory identity, proposing grid cells encode structure, and place cells form a conjunctive representation between sensory identity and structure (Fig 1a). This conjunctive representation forms a Hebbian memory, which bridges structure and identity, allowing the same structural code to be reused across environments that share statistics but differ in sensory experiences. We combine fast Hebbian learning of episodic memories, with gradient descent which slowly learns to extract statistics of these memories. Our network learns representations that mirror those found in the brain, with different entorhinal-like representations forming depending on transition statistics. We further present analyses of a remapping experiment [5], which support our model assumptions, showing place cells remap to locations consistent with a grid code, i.e. not randomly as previously thought.

The key conceptual novelties are as follows. **Neuroscience:** We found an interpretation of grid cells, place cells and remapping that offers a mechanistic understanding for the hippocampal involvement in generalisation of knowledge across domains. We provide a unifying framework for many entorhinal

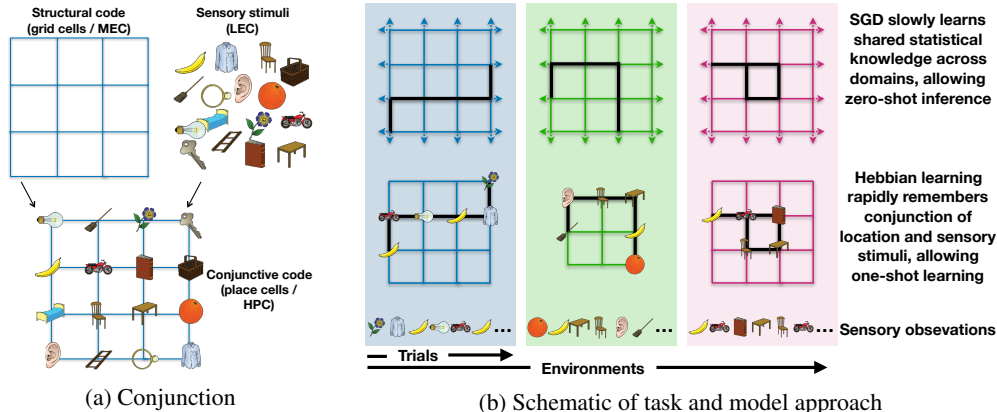

(a) Conjunction          (b) Schematic of task and model approach

Figure 1: (a) Separated structural and sensory representations combined in a conjunctive code. LEC/MEC: Lateral/Medial entorhinal cortex, HPC: Hippocampus. (b) Problem the model faces - extracting generalisable statistics across domains, while rapidly learning the map within domain.

cell types (grid cells, border cells, object vector cells) as building basis functions for transitions statistics. Our results suggest spatial representations found in the brain may be an instance of a more general coding mechanism organising knowledge across multiple domains. **Machine learning:** We have built a network where fast Hebbian learning interacts with slow statistical learning. This allows us to learn representations whereby memories not only stored in a Hebbian network for one-shot retrieval within domain, but also benefit from statistical knowledge that is shared across domains - allowing zero shot inference.

## 2   Related work

Concurrently developed papers discovered grid-like and/or place-like representations in ANNs [4, 10]. Neither paper uses memory or explains place cell phenomena. Both, however, use *supervised* learning in order to discover these representations, either supervising on actual $x, y$ coordinates [10] or ground truth place cells [4]. We use *unsupervised* learning, providing the network with only sensory observations and actions. This is information available to a biological agent, unlike ground truth spatial representations. We further propose a role for grid cells in generalisation, not just navigation.

Our modelling approach is simliar to [15 **?** ]. However, we choose our memory storage and addressing to be computationally biologically plausible (rather than using other types of differentiable memory more akin to RAM), as well as using hierarchical processing. This enables our model to discover representations that are useful for both navigation and addressing memories. We also are explicit in separating out the abstract structure of the space from any specific content (Fig 1a).

We follow a similar ideology to complementray learning systems [27] where the statistics of memories in hippocampus are extracted by cortex. We additionally propose that this learnt structural knowledge constrains hippocampal representations in new contexts, allowing reuse of learnt knowledge.

## 3   Model

We consider an agent passively moving on a 2D graph (Fig 1b), observing a non-unique sensory stimulus (e.g. an image) on each vertex. If the agent wishes to 'understand' its environment then it should maximise its model's probability of observing each stimulus. The agent is trained on many environments sharing the same structure, i.e. 2D graphs, however the stimulus distribution is different (each vertex is randomly assigned a stimulus). There are various approaches to this problem, however a generalisable one should exploit the underlying structure of the task - the 2D-ness of the space. One such approach is to have an abstract representation of space encoding relative locations, and then to place a memory of what stimulus was observed at that (relative) location. Since the agent understands where it is in space, this allows for accurate state predictions to previously visited nodes even if the agent has never travelled along that particular edge before (e.g. loop closure in Figs 1b pink and 2c). Although we consider 2D graphs to compare learned representations to those found in the brain, our approach is appropriate for generalising other stuctural/relational/conceptual knowledge [25].

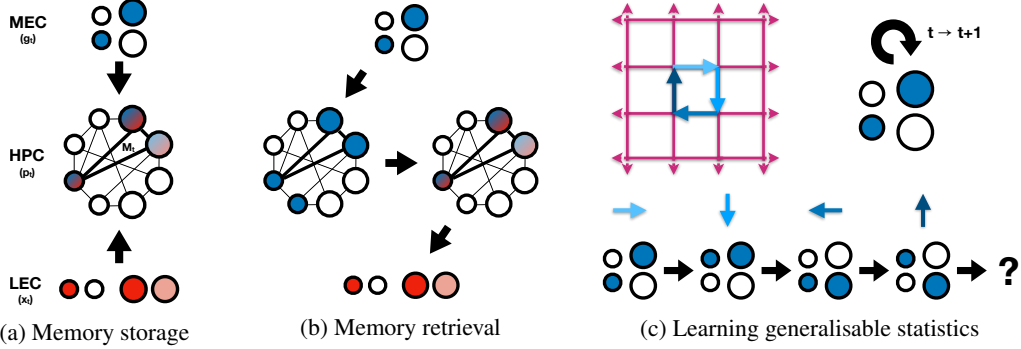

(a) Memory storage        (b) Memory retrieval        (c) Learning generalisable statistics

Figure 2: Learning good representations. Small/large circles: high/low frequency cells. Grid cells (MEC) need to create (a) conjunctive Hebbian memories (in HPC, weights $M_t$ between place cells) such that the same grid code can reinstate (b) the same memory via attractor dynamics. Grid cells are recurrent (c), and so must learn transition weights such that they have the same code when returning to a state (loop closure). This code must be general enough to work across many environments.

We propose grid cells as bases for constructing abstract representations of space, and place cell representations for the formation of fast episodic memories (Fig 2a). To link a stimulus to a given (relative) location, a memory should be a conjunction of abstract (relative) location and sensory stimulus, thus we propose place cells form a conjunctive representation between the sensorium and grid input (Figs 1a and 2a). This is consistent with experimental evidence [39, 22]. We posit that this is done hierarchically across spatial frequencies, such that the higher frequency statistics can be repeatedly used across space. This reduces the number of weights that need to be learnt. This proposition is consistent with the hierarchical scales observed across both grid cells [34] and place cells [21], and with the entorhinal cortex receiving sensory information in hierarchical temporal scales [37]. We consider grid cells to be recurrent through time, allowing predictive state transitions to occur via grid cells (Fig 2c). This is consistent with grid codes being a natural basis for navigation in 2D spaces [33, 8].

We view the hippocampal-entorhinal system as one that performs inference. Grid cells make inferences based on their previous estimate of location in abstract space (and optionally sensory information linked to previous locations via memory). Place cells, a conjunction between the sensory data and location in abstract space, are stored as memories. We consider sensory data, the item/object experience of a state, as coming from the 'what stream' via lateral entorhinal cortex. The grid cells in our model, are the 'where stream' coming from medial entorhinal cortex (Fig 2). Our hippocampal conjunctive memory links 'what' to 'where', such that when we revisit 'where' we remember 'what'.

## 3.1 Model summary

The model is a neural network and learns structure across tasks. We optimise end-to-end via backpropagation through time. The central (attractor) network employs Hebbian learning to rapidly remember the conjunction of location and sensory stimulus. A generative temporal model learns how to use the Hebbian memory most efficiently given the common statistics of transitions across worlds.

## 3.2 Generative model

We consider the agent to have a generative model (Fig 3a, schematic in Figs 2b, 2c) which factorises as $p_\theta\left(\mathbf{x}_{\leq T}, \mathbf{p}_{\leq T}, \mathbf{g}_{\leq T}\right) = \prod_{t=1}^{T} p_\theta\left(\mathbf{x}_t \mid \mathbf{p}_t\right) p_\theta\left(\mathbf{p}_t \mid \mathrm{M}_{t-1}, \mathbf{g}_t\right) p_\theta\left(\mathbf{g}_t \mid \mathbf{g}_{t-1}, \mathbf{a}_t\right)$ where observed variable $\mathbf{x}_t$ is the instantaneous sensory stimulus and latent variables $\mathbf{g}_t$ and $\mathbf{p}_t$ are grid and place cells. $\mathrm{M}_t$ represents the agent's memory composed from past place cell representations. $\mathbf{a}_t$ represents the current action - our version of *head-direction* cells [35]. $\theta$ are parameters of the generative model.

We now give concise, but intuitive descriptions of the model components. Expanded details in Supplementary Materials (SM). Sensory data $\mathbf{x}_t$ is a one-hot vector where each of its $n_s$ elements represent a sensory identity. We consider place and grid cells, $\mathbf{p}_t$ and $\mathbf{g}_t$ respectively, to come in different frequencies (hierarchies) indexed by superscript $f$. Though we already refer to these variables as grid and place cells, it is important to note that grid-ness and place-ness are not hard-coded - all representations are learned. $f(\cdots)$ denotes functions specific to the distribution in question.

**Grid cells.** To predict where we will be, we can transition from our current location based on our heading (i.e. path integration, schematic in Fig 2c). $p_\theta\left(\mathbf{g}_t \mid \mathbf{g}_{t-1}, \mathbf{a}_t\right)$ is a Gaussian transition probability density, with transitions taking the form $\mathbf{g}_t = f_{\mu_g}(\mathbf{g}_{t-1} + D_a\,\mathbf{g}_{t-1}) + \sigma \cdot \varepsilon_t$ with $\varepsilon_t \sim \mathcal{N}(0, \boldsymbol{I})$, $Vec[D_a] = f_D(\mathbf{a}_t)$ and $\sigma = f_{\sigma_g}(\mathbf{g}_{t-1})$. Connections in $D_a$ are from low frequency to the same or higher frequency only (or alternatively only within frequency). We separate into hierarchical scales so that high frequency statistics can be reused across lower frequency statistics, i.e. learning and knowledge is reused across space.

**Place cells.** $p_\theta\left(\mathbf{p}_t \mid \mathrm{M}_{t-1}, \mathbf{g}_t\right)$ is a Gaussian probability density for retrieving memories. Stored memories are extracted via an attractor network (Fig 2b) using $f_g(\mathbf{g}_t)$ as input - i.e. grid cells act as an index for memory extraction (Details in Section 3.4).

**Data.** We classify a stimulus identity using $p_\theta\left(\mathbf{x}_t \mid \mathbf{p}_t\right) \sim Cat\left(f_x(\mathbf{p}_t)\right)$.

### 3.3   Inference network

Due to the inclusion of memories, as well as other non-linearities, the posterior $p\left(\mathbf{g}_t, \mathbf{p}_t \mid \mathbf{x}_{\leq t}, \mathbf{a}_{\leq t}\right)$ is intractable - we therefore turn to approximate inference. To infer on this generative model, we make critical de-

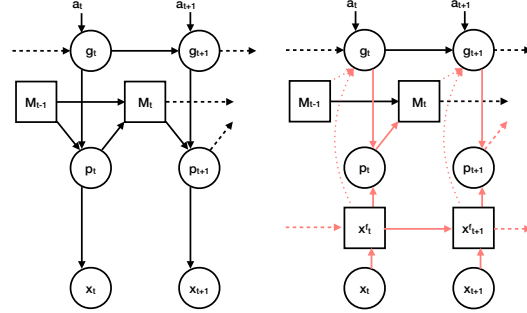

(a) Generative model    (b) Inference network

Figure 3: Circled/ boxed variables are stochastic/ deterministic. Red arrows indicate additional inference dependencies. Dashed arrows continue through time. Dotted arrow are optional as explained below.

cisions that respect our proposal of structural information separated from sensory information as well as respecting biological considerations. We use a recognition distribution that factorises as $q_\phi\left(\mathbf{g}_{\leq T}, \mathbf{p}_{\leq T} \mid \mathbf{x}_{\leq T}\right) = \prod_{t=1}^{T} q_\phi\left(\mathbf{g}_t \mid \mathbf{x}_{\leq t}, \mathrm{M}_{t-1}, \mathbf{g}_{t-1}\right) q_\phi\left(\mathbf{p}_t \mid \mathbf{x}_{\leq t}, \mathbf{g}_t\right)$. See Fig 3b for inference network schematic. $\phi$ denote parameters of the inference network. We learn $\theta$ and $\phi$, by maximising the ELBO with the variational autoencoder framework [20, 30] (details in SM).

**Place cells.** We treat these variables as a conjunction between sensorium and structural information from grid cells (Fig 2a). The sensorium is obtained by first compressing the immediate sensory data, $\mathbf{x}_t$, to $f_c(\mathbf{x}_t)$, after which it is filtered via exponential smoothing into different frequency bands, $\mathbf{x}_t^f$. After a normalisation step, each $\mathbf{x}_t^f$ is combined conjunctively with $\mathbf{g}_t^f$ to give the mean of the distribution $q_\phi\left(\mathbf{p}_t \mid \mathbf{x}_{\leq t}, \mathbf{g}_t\right)$. The separation into hierarchical scales helps to provide a unique code for each position, even if the same stimulus appears in several locations of one environment, since the surrounding stimuli, and therefore the lower frequency place cells, are likely to be different. Since the place cell representations form memories, one can utilise the hierarchical scales for memory retrieval. We note that although the exponential smoothing appears over-simplified, it approximates the Laplace transform with real coefficients. Cells of this nature have been discovered in LEC [37].

**Grid cells.** We factorise $q_\phi\left(\mathbf{g}_t \mid \mathbf{x}_{\leq t}, \mathrm{M}_{t-1}, \mathbf{g}_{t-1}\right)$ as $q_\phi\left(\mathbf{g}_t \mid \mathbf{g}_{t-1}, \mathbf{a}_t\right) q_\phi\left(\mathbf{g}_t \mid \mathbf{x}_{\leq t}, \mathrm{M}_{t-1}\right)$. To know where we are, we can path integrate ($q_\phi\left(\mathbf{g}_t \mid \mathbf{g}_{t-1}, \mathbf{a}_t\right)$ - equivalent to the generative distribution described above) as well as use sensory information that we may have seen previously ($q_\phi\left(\mathbf{g}_t \mid \mathbf{x}_{\leq t}, \mathrm{M}_{t-1}\right)$). The second distribution (optional addition) provides information on location given the sensorium. Since memories link location and sensorium, successfully retrieving a memory given sensory input allows us to refine our location estimate. Experimentally this improves training.

### 3.4   Hebbian memories

**Storage.** Memories of place cell representations are stored in Hebbian weights between place cells ($\mathrm{M}_t$ in Fig 2a), similar to [3]. We choose Hebbian learning, not only for its biological plausibility, but to also allow rapid learning when entering a new environment . We use the following learning rule to update the memory: $\mathrm{M}_t = \lambda\,\mathrm{M}_{t-1} + \eta(\mathbf{p}_t - \hat{\mathbf{p}}_t)(\mathbf{p}_t + \hat{\mathbf{p}}_t)^T$, where $\hat{\mathbf{p}}_t$ represents place cells generated from inferred grid cells. $\lambda$ and $\eta$ are the rate of forgetting and remembering respectively. Connections from high to low frequencies are set to zero, so that memories are retrieved hierarchically. We note than many other types of Hebbian rules work. In SM we describe changes to the learning rule if the additional distribution $q_\phi\left(\mathbf{g}_t \mid \mathbf{x}_{\leq t}, \mathrm{M}_{t-1}\right)$ is included for inference of grid cells.

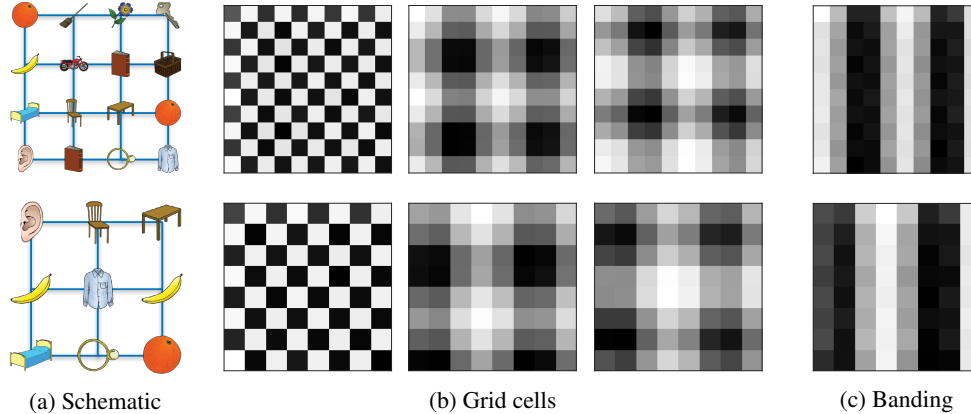

| (a) Schematic | (b) Grid cells | (c) Banding |

Figure 4: Top panel all one environment, bottom panels another environment. (a) Schematic of two environments (not actual size). (b) Same three grid cells from each environment. High frequency grid on left, lower frequency on the right. Same grid code used in environments of different sizes implies a general way of representing space, i.e. not just a template of each environment. These are square grids as we have chosen a four way embedding of actions and four connected space. (c) Banded cell.

**Retrieval.** To retrieve memories, similarly to [3], we use an attractor network of the form $\mathbf{h}_\tau = f_p\left(\alpha\mathbf{h}_{\tau-1} + \mathbf{M}_t\,\mathbf{h}_{\tau-1}\right)$, where $\tau$ is the iteration of the attractor network and $\alpha$ is a decay term. The input to the attractor, $\mathbf{h}_0$, is from the grid cells or sensorium (with their dimensions scaled appropriately) depending on whether memories are being retrieved for generative or inference purposes respectively. The output of the attractor is the retrieved memory (place cell code).

### 3.5 Model implications

We offer a solution to the problem of how structural codes are shared, via grid cells, to remapped place cells. Even with identical structure, since sensory stimuli across environments are different, the conjunctive code is different. Thus we believe that place cell remapping is not random, instead place cells are chosen that are consistent with both grid and sensory codes. This is a different notion to other remapping models [1], where random grid modular realignment produces a new set of place cells, and learning, that anchors these new representations, starts afresh in each environment. Our method allows for dramatically faster learning, as learnt structure can be re-used in new environments. In section 5, we present experimental evidence in concordance with our model.

In addition to offering a novel theory for place cell remapping, our model also provides an explanation for what determines place field sizes. Specifically, a given place cell will be active in the regions of space that are consistent with *both* with the grid representation (structure) received by that place cell and the sensory experience coded for by that place cell. It further offers explanation for why a given place cell may have multiple place fields within one environment, as there may be multiple locations where this consistency holds. Therefore our model offers a novel framework for designing experiments to manipulate place field sizes and locations, for example, based on simultaneously recorded grid cells and environmental cues.

We believe that using more biologically realistic computational mechanisms (e.g. Hebbian Memory, no LSTM) will facilitate further incorporation of neuroscience-inspired phenomena, such as successor representations or replay, which may be useful for building AI systems.

## 4 Model experiments

We show that by predicting sensory observations in environments that share structure, the model learns to generalise structural knowledge. This knowledge is represented similarly to spatial cells observed in the brain, suggesting these cells play a key role in generalisation. We further show our model exhibits fast (one-shot) learning and performs inference of unseen relationships. Although we presented a probabilistic formulation, best results were obtained when only considering the means of each distribution. Further implementation details in SM. We have taken a didactic approach to our model, thus we do not expect stellar model performance, nevertheless the model performs well.

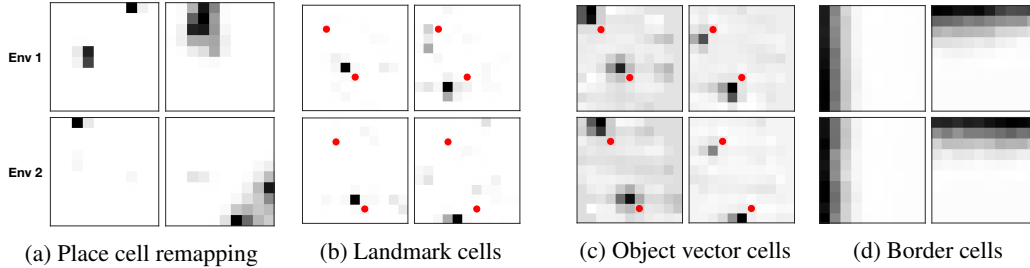

(a) Place cell remapping    (b) Landmark cells    (c) Object vector cells    (d) Border cells

Figure 5: Hippocampal cells (a, b) depend on sensory experience, whereas entorhinal cells (c, d) generalise over sensory experience. Example cells from two different environments (top/bottom). a) Place cells demonstrating remapping, as also observed in the brain. These are typical in the model. Left/right: High/low frequency cell. b) Hippocampal landmark cells fire at a specific distance and direction from objects, not generalising over object identities. c) Object vector cells, however, generalise both within and across environments. d) Border cells.

**Learned spatial representations.** We show the representations learned by our network in Fig 4 and 5 by plotting spatial activity maps of particular neurons. In Fig 4b we present grid cells. The top panel shows cells from one environment, and the bottom panels from a different and slightly smaller environment. We see that our network chooses to represent information in a grid-like pattern (square-grids as the statistics of our space is square). We can also observe spatial firing fields at different frequencies. Representations are consistent across environments, regardless of their size - thus we have a generalisable representation of 2D space, not just a template of a particular sized environment. Fig 4c shows banded cells from our model which are also observed in the brain alongside grid cells [24]. Further learned representations are shown in SM.

We observe the appearance of phases in the grid cells (middle and right panels of Fig 4b), i.e. we find grid representations that are shifted versions of each other, as in the brain [16]. The separation into different phases means that two conjunctive place cells that respond to the same stimulus, will not necessarily be active simultaneously - each cell will only be active when their corresponding grid phase is active. Thus one can uniquely code for the same stimulus in many different locations. Across two environments, a given stimulus may occur at the same grid phase but at a different location. Thus, due to their conjunctive nature, place cells may remap across environments, as in the brain. We show this in Fig 5a. Further learned place representations are shown in SM.

**Transition statistics determine basis functions.** By changing transition statistics, other entorhinal cell types are observed in our model. Encouraging the agent to spend more time near boundaries leads to the emergence of border cells [31] (Fig 5d). Biasing towards particular sensory experiences leads to the discovery of object vector cells [19] (Fig 5c). Similarly to experimental evidence, although these object vector cells in our 'grid' cell layer generalise over objects, the equivalent landmark cells [11] in our 'place' cell layer do not - they are object specific (Fig 5b). Our results suggest that the 'zoo' of different cell types found in entorhinal cortex may be viewed under a unified framework - summarising the common statistics of tasks into basis functions that can be flexibly combined depending on the particular structural constraints of the environment the animal/agent faces. After an initial guess of task structure, appropriate weighting of the bases can be inferred on-line (e.g. by sensory cues / performance) to parsimoniously describe the current task structure.

**One-shot learning.** We test whether the network can remember what it has just seen. We consider occasions when the agent stays still at a node for the first time, as a function of the number of times that node has previously been visited (Fig 6a). We see that the agent is able to predict at a high accuracy even if it has only just visited the node for the first time. This indicates we are able to do one-shot-learning with Hebbian memory, demonstrating our model can learn episodic memories.

**Zero-shot inference.** Having learned the structure of our space, we should be able to correctly predict previously visited nodes even if we approach from a non-traversed edge - i.e. infer a link in the graph on loop closure. We present such data in Fig 6b. We plot the prediction accuracy of such link inferences as a function of the fraction of the total nodes visited in the graph. We achieve considerably better than chance ($1/n_s = 0.02$) prediction, which remains stable throughout graph traversal. This shows that structural information is used for inferring unseen relationships.

**Long term memories.** Despite using BPTT truncated at 25 steps, we retain memories for much longer (Fig 6c), indicating our grid code allows efficient storage and retrieval of episodic memories.

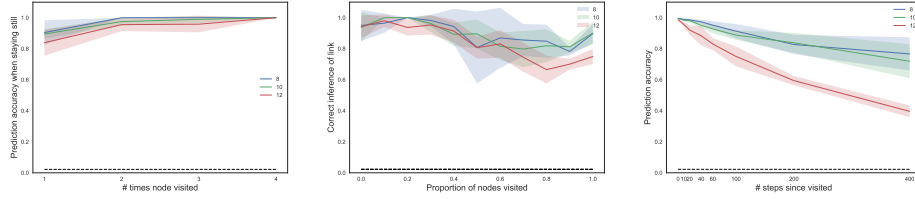

| (a) One-shot learning | (b) Zero-shot link inference | (c) Long term memories |

Figure 6: Prediction accuracy for different box widths. (a), (b) are previously unseen links only. Black dashed line is chance. (a) Attractor network is immediately able to retrieve Hebbian memories. (b) Unobserved graph links are inferred, implying the network has successfully learned and generalised structural knowledge. (c) Memories are successfully retrieved a long time after initial storage.

To reiterate, no representations are hard-coded. Place-like representations are learned in the attractor. Grid-like (and other entorhinal) representations are learned in the generative temporal model. These emerge from end-to-end training. These grid-like representations allow zero shot inference in new worlds demonstrating structural generalisation.

# 5 Analysis of data from a remapping experiment

Our framework predicts place cells and grid cells retain their relationship across environments, allowing generalisation of structure encoded by grid cells. We empirically test this prediction in data from a remapping experiment [5] where both place and grid cells were recorded from rats in two different environments. The environments were of the same dimensions (1m by 1m) but differed in their sensory (texture/visual/olfactory) cues so the animals could distinguish between them. Each of seven rats has recordings from both environments. Recordings on each day consist of five twenty-minute trials in the environments: the first and last trials in one environment and the intervening three trials in a second environment.

## 5.1 Methods

We test the prediction that a given place cell retains its relationship with a given grid cell across environments using two measures. First, whether grid cell activity at the position of peak place cell activity is correlated across environments (gridAtPlace), and second, whether the minimum distance between the peak place cell activity and a peak of grid cell activity is correlated across environments (minDist; normalised to corresponding grid scale). To account for potential confounds or biases (e.g. border effects, inaccurate peaks), we fit the recorded grid cell rate maps to an idealised grid cell equation [33], and use this ideal grid rate map to give grid cell firing rates and locations of grid peaks. Only grid cells with high grid scores ($> 0.8$) were used to ensure good ideal grid fits to the data, and we excluded grid cells with large scales ($> 50$cm), both computed as in [5]. Locations of place cell peaks were simply defined as the location of maximum activity in a given cell's rate map. To account for border effects, we removed place cells that had peaks close to borders ($< 10$cm from a border).

Our framework predicts a preserved relationship between place and grid cells of the same spatial scale (module). However, since we do not know the modules of the recorded cells, we can only expect a non-random relationship across the entire population. For each measure, we compute its value for every place cell-grid cell pair (from two trials). A correlation across trials is then performed on these values. To test the significance of this correlation and ensure it is not driven by bias in the data, we generate a null distribution by randomly shifting the place cell rate maps and recomputing the measures and their correlation across trials. We then examine where the correlation of the non-shuffled data lies relative to the null.

## 5.2 Results

We present analyses for both the gridAtPlace measure (Fig 7a) and the minDist measure (Fig 7b). The scatter plots show the correlation of a given measure across trials, where each point is a place cell-grid cell pair. The histogram plots show where this correlation (green line) lies relative to the null distribution of correlation coefficients. The p value is the proportion of the null distribution that is greater than the unshuffled correlation.

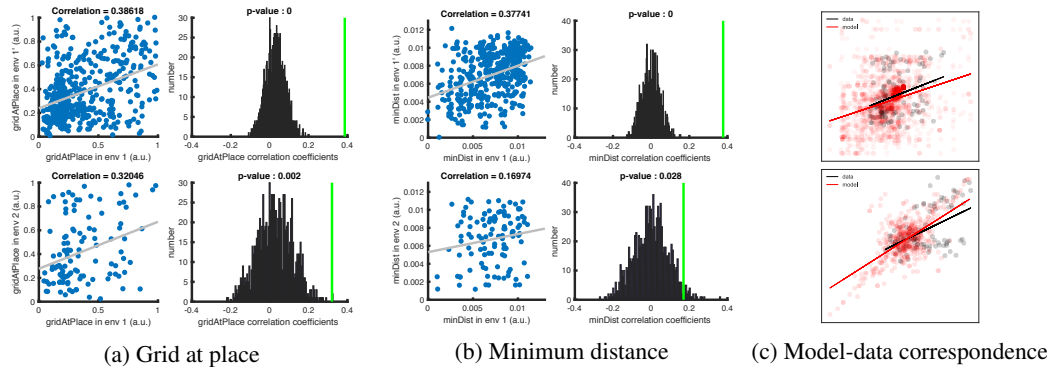

(a) Grid at place     (b) Minimum distance     (c) Model-data correspondence

Figure 7: (a), (b) Data analysis results: top panels are for within environment analyses, bottom panels across environment analyses. (c) Black/ Red: Data/ Model. Top: gridAtPlace across environments. Bottom: Scatter of elements of correlation matrices across environments.

As a sanity check, we first confirm these measures are significantly correlated *within* environments (i.e. across two visits to the same environment - trials 1 and 5), when the cell populations should not have remapped (see Fig 7a, top and 7b, top). We then test *across* environments (i.e. two different environments - trials 1 and 4), to asses whether our predicted non-random remapping relationship between grid and place cells exists (Fig 7a, bottom and 7b, bottom). Here we also find significant correlations for both measures for the 115 place cell-grid cell pairs. We note the gridAtPlace result holds across environments ($p < 0.005$) when not fitting an ideal grid and using a wide range of grid score cut-offs (minDist not calculated without the ideal grid due to inaccurate grid peaks). Finally performing the across environment gridAtPlace analysis with our *model* rate maps (Fig 7c top), we observe correlations of 0.3-0.35, which are consistent with that of the data.

To share structure, the relationship between grid cells should be preserved across environments. The grid cell correlation matrix is preserved (i.e. itself correlated) across environments ($p < 0.001$ from null), both in the data [5] as well as in our model (Fig 7c bottom). These results are consistent with the notion that grid cells encode generalisable structural knowledge.

These are the first analyses demonstrating non-random place cell remapping based on neural activity, and provide evidence for a key prediction of our model: that place cells, despite remapping across environments, retain their relationship with grid cells.

# 6  Conclusions

We proposed a mechanism for generalisation of structure inspired by the hippocampal-entorhinal system. We proposed that one can generalise state-space statistics via explicit separation of structure and stimuli, while using a conjunctive representation with fast memory to link the two. We proposed that spatial hierarchies are utilised to allow for an efficient combinatorial code. We have shown that hierarchical grid-like and place-like representations emerge naturally from our model in a purely unsupervised learning setting. We have shown that these representations are effective at both generalising the state-space (zero-shot link inference), but also for hierarchical memory addressing. We have proposed that entorhinal cortex provides a basis set for describing the current transition structure, unifying many entorhinal cell types. We have suggested that spatial coding is just one instance of a broader framework organising knowledge. Our framework incorporates numerous phenomena or functions ascribed to the hippocampal formation (spatial cognition and representations, conceptual knowledge representation, hierarchical representations, episodic memory, inference, and generalisation). We have also presented experimental evidence that demonstrates grid and place cells retain their relationships across environment, which supports our model assumptions. We hope that this work can provide new insights that will allow for advances in AI, as well as providing new predictions, constraints and understanding in neuroscience.

# 7  Author Contributions

JCRW developed the model, performed simulations and drafted paper. CB collected data. JCRW, THM analysed data. JCRW, THM, SM, TEJB conceived project and edited paper. TEJB supervised.

# 8 Acknowledgements

We acknowledge funding from a Wellcome Trust Senior Research Fellowship (WT104765MA) together with a James S. McDonnell Foundation Award (JSMF220020372) to TEJB, MRC scholarship to THM, and an EPSRC scholarship to JCRW.

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
