[Supplementary Material · NeurIPS_supplementary_material_v2.pdf]

# Generalisation of structural knowledge in the hippocampal-entorhinal system: Supplementary Material

**James C.R. Whittington\***
University of Oxford, UK
james.whittington@magd.ox.ac.uk

**Timothy H. Muller\***
University of Oxford, UK
timothymuller127@gmail.com

**Shirley Mark**
University College London, UK
s.mark@ucl.ac.uk

**Caswell Barry**
University College London, UK
caswell.barry@ucl.ac.uk

**Timothy E.J. Behrens**
University of Oxford, UK
behrens@fmrib.ox.ac.uk

## 1    Additional model details

We denote a layer of activations with vector notation e.g. $\mathbf{p}_t$ or $\mathbf{p}_t^f$ for a given frequency. Otherwise variables with subscripts $s$ and/ or $j$ represent elements of the corresponding vector e.g. $p_{t,s,j}^f$ - a place cell in frequency $f$ of (compressed) sensory preference $s$ and of grid preference $j$. We use $w$ to denote scalar weights, $W$ for matrices and $b$ for biases. The sensory data $\mathbf{x}_t$ is a one-hot vector where each of its $n_s$ elements $x_{t,s}$ represent a particular sensory identity $s$. This sensory data is later compressed to dimension $n_{s*}$. We consider place and grid cells, $\mathbf{p}_t$ and $\mathbf{g}_t$ respectively, to come in different frequencies indexed by the superscript $f$. A grid cell in a given frequency is denoted by $g_{t,j}^f$, where the index $j$ is over the number of grid cells in that frequency. A place cell also has a particular (compressed) sensory preference - we denote this by $p_{t,s,j}^f$ where the index $j$ is over the number of 'phases' in that frequency ($n^f$), i.e. there are $n^f n_{s*}$ place cells for frequency $f$. Note there may be more than $n^f$ grid cells per frequency due to the function $f_g(\mathbf{g}_t)$ (see below).

### 1.1    Generative model

**Grid cell generation.** We chose the function $f_{\mu_g}(\cdots)$ to be linear, but thresholded at $\pm 1$. $f_{\sigma_g}(\cdots)$ is a simple MLP and $f_D(\cdots)$ similarly.

**Place cell generation.** $p_\theta\left(\mathbf{p}_t \mid M_{t-1}, \mathbf{g}_t\right) = \mathcal{N}(\mu(M_{t-1}, \mathbf{g}_t), \sigma(M_{t-1}, \mathbf{g}_t)$ where $\mu(M_{t-1}, \mathbf{g}_t)$ is the retrieved memory, and $\sigma$ is a simple MLP of $\mu$. The input to the attractor network, $f_g(\mathbf{g}_t)$, we define as a subset of $\mathbf{g}_t$ repeated appropriately to have the correct dimensions (for each frequency).

**Data generation.** $p_\theta\left(\mathbf{x}_t \mid \mathbf{p}_t\right)$ is a categorical distribution. We define $f_x(\cdots)$ to be $softmax\left(f_{c*}\left(\sum_f^{f^*} w_x^f \sum_j p_{t,s,j}^f + b_x\right)\right)$, summing over 'phases', where $w_x^f$ is a learnable parameter for each frequency and $f_{c*}$ is a MLP for 'decompressing' into the correct input dimensions. We choose $f^*$ to be 0 (i.e. only include highest frequency).

## 1.2 Inference network

**Place cell inference.** $q_\phi\left(\mathbf{p}_t \mid \mathbf{x}_{\leq t}, \mathbf{g}_t\right)$. We describe the process to obtain the mean of this distribution first. We treat these neurons as a conjunction between sensorium and structural information from the grid cells. To obtain the sensorium we first compress our one-hot encoding of instantaneous data $f_c(\cdots)$, which we choose to be a two-hot encoding (or a learnable encoding). We then exponentially smooth this with $\mathbf{x}_t^f = \left(1 - \alpha^f\right)\mathbf{x}_{t-1}^f + \alpha^f f_c(\mathbf{x}_t)$ into different temporal scales using learnable smoothing constants $\alpha^f$. We then normalise each frequency with $f_n(\mathbf{x}_t^f)$, where $f_n()$ demeans then applies a relu followed by unit normalisation. We combine conjunctively with $\mu_t = f_p(\tilde{\mathbf{g}}_t \cdot \tilde{\mathbf{x}}_t)$ where $\tilde{g}_{t,s,j}^f = f_g(g)_{t,j}^f$ and $\tilde{x}_{t,s,j}^f = w_p^f f_n(\mathbf{x}_t^f)_s$ i.e. repeated $n_{s^*}$ and tiled $n^f$ times respectively to have the correct dimensions. The distribution's variance, $\sigma$, is given by a MLP with input $[f_n(\mathbf{x}_t^f), \mathbf{g}_t^f]$. We choose the $f_p$ to be a *leaky relu* to ensure the only neurons active are those 'consistent' with both the sensorium and the structural information. This also sparsifies our memories and prevents interference.

**Grid cell inference.** We describe the optional additional distribution $q_\phi\left(\mathbf{g}_t \mid \mathbf{x}_{\leq t}, M_{t-1}\right)$ further. It provides information on location from the current sensorium. Since memories are a conjunction between location and sensorium, a memory contains information regarding location. We use $\tilde{\mathbf{x}}_t$ as the input to the attractor network to retrieve the memory associated with the current sensorium. We use a, per frequency, MLP from the retrieved memory to give the mean of the distribution. The variance of the distribution is a function of the length of the retrieved memory, as well as how well the retrieved memory is able to reproduce the sensorium, i.e. if we are able to successfully retrieve a memory, we can be more confident that our memory is informative on current location. This factored distribution is a Gaussian with a precision weighted mean - i.e. we refine our generated location estimate with sensory information.

## 1.3 Hebbian memories

Each time the agent enters a new environment, the Hebbian memory, $M_t$, is reset to be empty (all weights zero). The exact Hebbian learning rule we choose is somewhat arbitrary, in that there are many other types of Hebbian learning rules which we found to be effective. Some other examples are $M_t = \lambda M_{t-1} + \eta(\mathbf{p}_t^i - \mathbf{p}_t^g)\tilde{\mathbf{g}}_t^T$ or $M_t = \lambda M_{t-1} + \eta(\mathbf{p}_t^i \mathbf{p}_t^{iT} - \mathbf{p}_t^g \mathbf{p}_t^{gT})$.

When using the sensorium to constrain the grid code, we can either use the same memory matrix as the generative case (as the brain presumably does), or we can use a separate memory matrix. Best results (and those presented) were when two separate matrices were used. We used the following learning rule for the inference based matrix: $M_t^* = \lambda M_{t-1}^* + \eta(\mathbf{p}_t^i - \mathbf{p}_t^x)(\mathbf{p}_t^i + \mathbf{p}_t^x)^T$, where $\mathbf{p}_t^x$ is the retrieved memory with the sensorium as input to the attractor. This second matrix did not have any restrictions on its connectivity.

## 1.4 Training

We wish to learn the parameters for both the generative model and inference network, $\theta$ and $\phi$, by maximising the ELBO, a lower bound on $\ln p_\theta\left(\mathbf{x}_{\leq T}\right)$. Following [4] (Section 5), we obtain a free energy $\mathcal{F} = \sum_{t=1}^T \mathbb{E}_{q_\phi\left(\mathbf{g}_{<t}, \mathbf{p}_{<t} \mid \mathbf{x}_{<t}\right)}[J_t]$, with $J_t = \mathbb{E}_{q_\phi(\ldots)}[\ln p_\theta\left(\mathbf{x}_t \mid \mathbf{p}_t\right) + \ln \frac{p_\theta(\mathbf{p}_t \mid M_{t-1}, \mathbf{g}_t)}{q_\phi\left(\mathbf{p}_t \mid \mathbf{x}_{\leq t}, \mathbf{g}_t\right)} + \ln \frac{p_\theta\left(\mathbf{g}_t \mid \mathbf{g}_{t-1}\right)}{q_\phi\left(\mathbf{g}_t \mid \mathbf{x}_{\leq t}, M_{t-1}, \mathbf{g}_{t-1}\right)}]$ as a *per time-step* free energy. We use the variational autoencoder framework [6, 7] to optimise this generative temporal model.

## 2 Implementation details

Although we have presented a Bayesian formulation, best results (those presented) were obtained by using a network of the identical architecture, however only using the means of the above distributions - i.e. not sampling from the distributions. We use the following surrogate loss function: $L_{total} = \sum_t L_{x_t} + L_{g_t} + L_{p_t}$ with $L_{x_t}$ being a cross entropy loss, and $L_{p_t}$ and $L_{g_t}$ are squared error losses between 'inferred' and 'generated' variables - in an equivalent way to the Bayesian energy function. We augment with a next time-step prediction loss, as well as a prediction loss from the inferred grid

cells. An additional squared error loss between the inferred memory and the retrieved memory given sensory input is used, should that module be included. We note that, like [3], a higher ratio of grid to band cells is observed if additional l2 regularisation of grid cell activity is used.

We use backpropagation through time truncated to 25 steps, and optimise with ADAM [5] with a learning rate that is annealed from $1e-3$ to $1e-4$. We use $n_s = 45$, $n_{s^*} = 10$ and 5 different frequencies, with $n^f$ as $[10, 10, 8, 6, 6]$. Our environments are square with possible widths $[8, 10, 12]$. The agent changes to a completely new environment after a certain number of steps ($\sim$ 2000-5000). The agent has a slight bias for straight paths to facilitate equal exploration. $\lambda$ and $\eta$ are set to $0.9999$ and $0.5$ respectively. $\mathbf{a}_t$ is a direction signal, where the agent can move, up, down, left, right or stay still. Initially we down-weight costs not associated with prediction. We do not train on vertices that the agent has not seen before. Code will be made available at `http://www.github.com/djcrw/generalising-structural-knowledge`.

For all simulations presented above, we use the additional memory module in grid cell inference. We do so using two separate memory matrices. For simulations involving object vector cells, we also use an extra factored distribution in grid cell inference: $q_\phi(\mathbf{g}_t \mid s_t)$ - where $s_t$ is an indicator telling the network if it is at the location of a 'shiny' state. We also remove $\mathbf{a}_t$ from the generative model, but it is still included in the inference network - i.e. two different distributions for grid transitions, one with direction information (inference) and one without (generative). We do this so that the generative model can more easily capture the true underlying transition statistics.

Typically after $200 - 300$ environments, the agent has fully learned the structure. This equates to $\sim 50000$ gradient updates. There are many simple extensions to improve performance, at the expense of computation, e.g. hyper-parameter tuning, normalisation for the attractor ([1, 2]).

# 3 Grid cell representations

Here we show learned grid cells. Note the distinct frequency modules. These cells are not all from the same model or environment size.

Figure 1: Higher frequency grid cells

Figure 2: Middle frequency grid cells

Figure 3: Lower frequency grid cells

# 4 Place cell representations

Here we show learned place cells. Note the distinct frequency modules. These cells are not all from the same model or environment size.

Figure 4: Higher frequency place cells

Figure 5: Middle frequency place cells

Figure 6: Lower frequency place cells

## 5 Derivation of variational lower bound

We follow the derivation from [4]. Exploiting Jensen's inequality, we can re-write as the following

$$\ln p_\theta \left( \mathbf{x}_{\leq t} \right) \geq \underset{q_\phi \left( \mathbf{p}_{\leq t}, \mathbf{g}_{\leq t} | \mathbf{x}_{\leq t} \right)}{\mathbb{E}} \ln \frac{p_\theta \left( \mathbf{x}_{\leq t}, \mathbf{p}_{\leq t}, \mathbf{g}_{\leq t} \right)}{q_\phi \left( \mathbf{p}_{\leq t}, \mathbf{g}_{\leq t} \mid \mathbf{x}_{\leq t} \right)}$$

Should we factorise both our generative and recognition distribution temporally as follows (we use the specific distributions from the paper later)

$$p_\theta \left( \mathbf{x}_{\leq t}, \mathbf{p}_{\leq t}, \mathbf{g}_{\leq t} \right) = \prod_{t=1}^{T} p_\theta \left( \mathbf{x}_t \mid \mathbf{x}_{<t}, \mathbf{p}_{\leq t}, \mathbf{g}_{\leq t} \right) p_\theta \left( \mathbf{p}_t \mid \mathbf{x}_{<t}, \mathbf{p}_{<t}, \mathbf{g}_{\leq t} \right) p_\theta \left( \mathbf{g}_t \mid \mathbf{x}_{<t}, \mathbf{p}_{<t}, \mathbf{g}_{<t} \right)$$

$$q_\phi \left( \mathbf{p}_{\leq t}, \mathbf{g}_{\leq t} \mid \mathbf{x}_{\leq t} \right) = \prod_{t=1}^{T} q_\phi \left( \mathbf{p}_t, \mathbf{g}_t \mid \mathbf{x}_{\leq t}, \mathbf{p}_{<t}, \mathbf{g}_{<t} \right)$$

We can then write things as the following

$$\ln p_\theta \left( \mathbf{x}_{\leq t} \right) \geq \underset{q_\phi \left( \mathbf{p}_{\leq t}, \mathbf{g}_{\leq t} | \mathbf{x}_{\leq t} \right)}{\mathbb{E}} \sum_{t=1}^{T} J_t$$

Where

$$J_t = \ln \frac{p_\theta \left( \mathbf{x}_t \mid \mathbf{x}_{<t}, \mathbf{p}_{\leq t}, \mathbf{g}_{\leq t} \right) p_\theta \left( \mathbf{p}_t \mid \mathbf{x}_{<t}, \mathbf{p}_{<t}, \mathbf{g}_{\leq t} \right) p_\theta \left( \mathbf{g}_t \mid \mathbf{x}_{<t}, \mathbf{p}_{<t}, \mathbf{g}_{<t} \right)}{q_\phi \left( \mathbf{p}_t, \mathbf{g}_t \mid \mathbf{x}_{\leq t}, \mathbf{p}_{<t}, \mathbf{g}_{<t} \right)}$$

Thus

$$\ln p_\theta \left( \mathbf{x}_{\leq t} \right) \geq \underset{q_\phi \left( \mathbf{p}_{\leq t}, \mathbf{g}_{\leq t} | \mathbf{x}_{\leq t} \right)}{\mathbb{E}} \sum_{t=1}^{T} J_t$$

$$= \int q_\phi \left( \mathbf{p}_t, \mathbf{g}_t \mid \mathbf{x}_1 \right) \int \dots \int q_\phi \left( \mathbf{p}_T, \mathbf{g}_T \mid \mathbf{x}_{\leq T}, \mathbf{p}_{<T}, \mathbf{g}_{<T} \right) \sum_{t=1}^{T} J_t$$

Since $J_t$ is not a function of elements from the set $\{ \mathbf{p}_{t+1}, \mathbf{g}_{t+1}, \mathbf{p}_{t+2}, \mathbf{g}_{t+2} \dots \mathbf{p}_T, \mathbf{g}_T \}$, we can rewrite the above equation as the following:

$$= \int q_\phi \left( \mathbf{p}_t, \mathbf{g}_t \mid \mathbf{x}_1 \right) J_1 \int q_\phi \left( \mathbf{p}_2, \mathbf{g}_2 \mid \mathbf{x}_{\leq 2}, \mathbf{p}_1, \mathbf{g}_1 \right) \dots \int q_\phi \left( \mathbf{p}_T, \mathbf{g}_T \mid \mathbf{x}_{\leq T}, \mathbf{p}_{<T}, \mathbf{g}_{<T} \right)$$

$$+ \int q_\phi \left( \mathbf{p}_t, \mathbf{g}_t \mid \mathbf{x}_1 \right) \int q_\phi \left( \mathbf{p}_2, \mathbf{g}_2 \mid \mathbf{x}_{\leq 2}, \mathbf{p}_1, \mathbf{g}_1 \right) J_2 \dots \int q_\phi \left( \mathbf{p}_T, \mathbf{g}_T \mid \mathbf{x}_{\leq T}, \mathbf{p}_{<T}, \mathbf{g}_{<T} \right)$$

$$+ \dots$$

$$+ \int q_\phi \left( \mathbf{p}_t, \mathbf{g}_t \mid \mathbf{x}_1 \right) \int q_\phi \left( \mathbf{p}_2, \mathbf{g}_2 \mid \mathbf{x}_{\leq 2}, \mathbf{p}_1, \mathbf{g}_1 \right) \dots \int q_\phi \left( \mathbf{p}_T, \mathbf{g}_T \mid \mathbf{x}_{\leq T}, \mathbf{p}_{<T}, \mathbf{g}_{<T} \right) J_T$$

All inner integrals integrate to 1, and so we are left with the following:

$$\mathcal{F} = \sum_{t=1}^{T} \underset{\prod_{\tau=1}^{t} q_\phi \left( \mathbf{p}_\tau, \mathbf{g}_\tau | \mathbf{x}_{\leq \tau}, \mathbf{p}_{<\tau}, \mathbf{g}_{<\tau} \right)}{\mathbb{E}} [J_t]$$

This can all we rewritten as:

$$\mathcal{F} = \sum_{t=1}^{T} \mathbb{E}_{\prod_{\tau=1}^{t-1} q_\phi(\mathbf{p}_\tau, \mathbf{g}_\tau | \mathbf{x}_{\leq\tau}, \mathbf{p}_{<\tau}, \mathbf{g}_{<\tau})} \Big[ \mathbb{E}_{q_\phi(\mathbf{p}_t, \mathbf{g}_t | \mathbf{x}_{\leq t}, \mathbf{p}_{<t}, \mathbf{g}_{<t})} [\ln p_\theta(\mathbf{x}_t \mid \mathbf{x}_{<t}, \mathbf{p}_{\leq t}, \mathbf{g}_{\leq t})$$
$$+ \ln p_\theta(\mathbf{p}_t \mid \mathbf{x}_{<t}, \mathbf{p}_{<t}, \mathbf{g}_{\leq t})$$
$$+ \ln p_\theta(\mathbf{g}_t \mid \mathbf{x}_{<t}, \mathbf{p}_{<t}, \mathbf{g}_{<t})$$
$$- \ln q_\phi(\mathbf{p}_t, \mathbf{g}_t \mid \mathbf{x}_{\leq t}, \mathbf{p}_{<t}, \mathbf{g}_{<t})]]$$

We can see that this is now an per time-step cost function that we can optimise. We now use add in our choice of distributions. First our generative distribution:

$$q_\phi(\mathbf{p}_t, \mathbf{g}_t \mid \mathbf{x}_{\leq t}, \mathbf{p}_{<t}, \mathbf{g}_{<t}) = q_\phi(\mathbf{p}_t \mid \mathbf{x}_{\leq t}, \mathbf{g}_t) \, q_\phi(\mathbf{g}_t \mid \mathbf{x}_{\leq t}, M_{t-1}, \mathbf{g}_{t-1})$$

and now our recognition distribution:

$$p_\theta(\mathbf{x}_{\leq t}, \mathbf{p}_{\leq t}, \mathbf{g}_{\leq t}) = p_\theta(\mathbf{x}_t \mid \mathbf{p}_t) \, p_\theta(\mathbf{p}_t \mid \mathbf{g}_t, M_{t-1}) \, p_\theta(\mathbf{g}_t \mid \mathbf{g}_{t-1})$$

With $M_{t-1}$ being the memory composed of previous $\mathbf{p}_t$ representations (stored in synaptic weights).

We can now simplify to the following:

$$\mathcal{F} = \sum_{t=1}^{T} \mathbb{E}_{\prod_{\tau=1}^{t-1} q_\phi(\mathbf{p}_\tau, \mathbf{g}_\tau | \mathbf{x}_{\leq\tau}, \mathbf{p}_{<\tau}, \mathbf{g}_{<\tau})} \Big[$$
$$+ \mathbb{E}_{q_\phi(\mathbf{p}_t, \mathbf{g}_t | \mathbf{x}_{\leq t}, \mathbf{p}_{<t}, \mathbf{g}_{<t})} [\ln p_\theta(\mathbf{x}_t \mid \mathbf{p}_t)]$$
$$- \mathbb{E}_{q_\phi(\mathbf{g}_t | \mathbf{x}_{\leq t}, M_{t-1}, \mathbf{g}_{t-1})} D_{\mathrm{KL}}(q_\phi(\mathbf{p}_t \mid \mathbf{x}_{\leq t}, \mathbf{g}_t) \, \| \, p_\theta(\mathbf{p}_t \mid \mathbf{x}_{<t}, \mathbf{g}_t))$$
$$- D_{\mathrm{KL}}(p_\theta(\mathbf{g}_t \mid \mathbf{x}_{<t}, M_{t-1}, \mathbf{g}_{t-1}) \, \| \, q_\phi(\mathbf{g}_t \mid \mathbf{x}_{\leq t}, M_{t-1}, \mathbf{g}_{t-1}))]]$$