[Reviews · NeurIPS 2018]

Reviewer 1



It is an interesting study that tackles one of very important questions in computational neuroscience - how generalisation across stimuli and environments is achieved. It is similar to the concept of schemas, which are thought to primarily rely on frontal cortical areas. In this particular case the focus is on entorhinal grid cells and hippocampal place cells, which authors assert code for the environment and conjunction of environment and stimulus respectively. The authors present a computational model that aims to address the question of whether place cells encode a conjunctive outcome of environment and stimulus representations. It is an interesting hypothesis, which if shown convincingly, would be a major breakthrough in neuroscience. However, this study suffers from lack of clarity in methodological descriptions as well as from seemingly arbitrary choices in analyses, which at least in the current version does not convincingly show that the stated hypothesis is supported by data. Both of these aspects should be improved for the paper to be acceptable to NIPS. More specifically, - it is clear (and supported by data) that grid cells have different frequencies (that go along anatomical gradients); however, for place cells the meaning of frequency is more ambiguous - is there evidence of similar anatomical gradients (based on projections from EC) of place field radii, which may suggest the support of frequencies for place cells as well as the actual hypothesis of conjunctive coding - it's not exactly clear which exactly formulations and equations are used to simulate the model & learning - Bayesian, Hebbian or BPTT - the latter being considerably less biologically realistic. I would suggest to move theoretical background to the appendix and instead present model description and simulation clearly in the main text, including all steps and their justification (including of some seemingly arbitrary choices) - it's not clear how exactly grid cells (and place cells) result from model constraints (or are they more hand-designed?) - considering noisiness of biological place cells, it doesn't seem very reasonable to consider peak activation as their centre - wouldn't it be better to fit a Gaussian and consider its centre? - Why are trials 1 vs 4 selected to evaluate the match between different environments? To avoid bias, it would be a lot better to include all pairwise comparisons of (1,5) vs (2,3,4) - In presenting the "within environment" comparison it would be equally interesting to see if comparing 2 vs 3 vs 4 lead to similar results as 1 vs 5, and if not, why not? - Correlations between measures seem a bit too low to support the claim regarding the relationship between grid and place cells in remapping (even if they are statistically significant). Mindist measure seems considerably smaller between environments than within the same environment (even if marginally significant). - Statistics in 7b seem to indicate a considerable mismatch between the experiment and model prediction - Finally what is meant by "anonymous authors" for ref. 3? - There are also many typos that should be corrected. --- I thank the authors for their rebuttal. As a result, paper clarity is going to be improved. However, I'm still not very convinced about biological plausibility of network structure and the way it's trained. Furthermore, some of my technical concerns (e.g. why 2-3-4 are not used for within environment comparison, as 1-5 are, or why novelty implies that only #4 should be used) have not been convincingly addressed either. Therefore, I do not change my score. I believe this story holds great potential but I'm not convinced yet.

Reviewer 2



Quality: This paper has a number of appealing features. There is a novel model of how the brain learns relational structure. The model consists of both an interesting probabilistic generative model as well as a neural recognition model, which together represent a plausible idea of how the grid cell and place cell representational systems cooperate to represent structured knowledge. The model makes a novel prediction: that there should be a consistent relationship between place cell and grid cell tunings across remappings between different environments. Then the paper presents an analysis of neural data claiming to show that, indeed, this prediction is borne out. On these grounds alone I lean towards acceptance. However, several considerations prevent me from giving it a higher score. I would be interested to hear from the authors on these points during the rebuttal phase and am open to changing my score. I confess I am not an expert in this area, but I have long found the topic interesting from a distance and suspect that the authors' basic approach is on target. Issues I would like to see addressed in a revision: * The simulations in Section 4 and illustated in Figure 4 seem highly abstract, and not closely to connected to the actual phenomena of grid cells that we see in the brain. Aren't the grids supposed to be hexagonal, rather than checkerboard-like as in Fig 3b? * The analyses of neural data seem reasonable to me, but pretty coarse-grained. Is this really the best one can do in linking up the model to neural data? Is this an in-principle limitation given the current data the authors analyze, or a51re there experiments one could do to establish a tighter link? * The figure captions were less helpful than I would have liked. I realize space is tight, but could you explain more clearly what is going on in Figures 4-7 and how they support the argument you are making? Clarity: The paper is reasonably clear apart a few typos. Originality: To my knowledge, both the model and the experimental data analysis are novel. But I am far from an expert. Significance: To the extent that this model and its key predictions are well supported by experimental data, the paper's significance is very high. Not being an expert I couldn't assess this well, but I would not be surprised to see a paper of this sort appearing in a high-impact neuroscience journal such as Nature Neuroscience or Neuron. Maybe for reasons of space limitation, or maybe because NIPS encourages submission of very recent, novel, but less than fully baked work, I felt that the argument presented here was not as strong as one would expect to see in a journal like Nature Neuro. Yet the work feels important in that way. If the paper is not accepted to NIPS, I would encourage the authors to refine it and submit it to a high-impact neuroscience journal. --------- Update after reading authors' response: Thanks to the authors for a thorough and thoughtful response. It was too compact for me to get very much out of, but I feel like they addressed several of my main concerns in a positive way, so I have raised my score accordingly. If the paper is not accepted at NIPS, I would urge the authors to submit to a neuroscience or computational neuroscience journal. The papers at NIPS are too short, and the review-response process too low-bandwidth, to do justice to this kind of work.

Reviewer 3



This paper proposes a computational model to investigate the relationship between place cells and grid cells. Overall, I have an extreme difficulty to follow what is actually done and in the current state the results seem impossible to reproduce. My difficulties are mostly due to vague statements, while concise and precise definition (without jargon) of the methods is missing. In addition the authors provide very little evidence that their approach is doing what it is supposed to do. Finally, I also have difficulty to understand the main claim of the paper. Details comments follow: Methods: Overall I have very little understanding and confidence in what is actually done. Key elements of the model are described with vague sentences such as line 134: “Following [14], we end up with the following per time-step function to optimise, for which we can use the reparameterisation trick and variational autoencoders.” If you use a variational autoencoder please state clearly the architecture in one single paragraph. Multiple statements about the network are scatter across several pages. I ask myself the following questions (among others): - What is the circle symbol on line 169 - What are a_t’s in Figure 2 - Where the concept of frequency come from? How quantities at a given frequency are computed - What is there two place cell layers (one in “inference arm” one in “generative arm”), why using an attractor network here? Is it based on the literature? Where can I get information about the properties of this chosen network. - Why state transitions on line 188 are Gaussian distributed (I thought space was discretized). - What does the schematic in Fig 3 (a) represent (compared to variables introduced in the text)? Results Only the spatial activation maps of few cells is provided. From that the authors claim there is evidence of place cell “remapping”. While remapping as a relatively clear meaning from an experimental neuroscience perspective, it is unclear to me what is meant in the context of these experiments, how it is quantified, and whether the result is non-trivial. I understand there is learning performed when switch from an environment to the other, so to some extend changes should occur in the synaptic weights (can the authors provide information about that?). Whether these changes correspond to a specific “remapping” of one grid cell to another (I assume this is what is meant) would require a detailed analysis of the associated synaptic weight. Moreover, given it was specifically chosen (judging from Figure 2) that place cells simply extract information from grid cells, isn’t such conclusion trivial? Can the authors propose a reasonable alternative hypothesis? I assume I am missing something, and this misunderstanding extend to the neural data analysis results. It is claimed on line 313 “Our framework predicts a preserved relationship between place and grid cells of the same spatial scale (module).” Do the authors provide evidence of that in there model, or is it and assumptions? How do they articulate the concept of “preserved relationship” with the concept of “remapping” between place and grid cells?

Reviewer 4



The authors present a model of grid cell – place cell interactions and apply it to the question of how associative (place-scene, place-object encounter) memories for separate environments and for separate locations in a single environment can be linked through a stable grid cell map like that of the medial entorhinal cortex in rats. Linked between-environment memories and linking in the form of predictions for upcoming changes in spatial and sensory context associated with locomotion through a single environment are achieved through mapping of locations within a grid network. One key to this is that the grid network is “universal” in the sense that the spatial firing relationships between members are constant. The larger meaning(s) of such universality in representation has not been adequately addressed in the literature. The authors test the model’s ability to predict the upcoming sensory context during locomotion for locations that, as yet, had not been visited during a given session of exploration. They also test the ability of the model to achieve “one-shot” learning. The performance of the model is strong in both respects. The effort and results are very original in attacking the problem of generalization using place-cell, grid cell network interactions. The model and its dynamics are clearly presented. The work represents an important contribution to the study of learning and memory processes as they relate to representation. It is well-positioned to inspire further work that takes the study of grid cells beyond the mapping of environment space. A few points that might enhance the presentation: 1) The authors might give some consideration, however short, to the issue of how variation in environmental border geometry, scale, and distal cue similarities would affect linking of place cell representations through the grid cell network. 2) The authors repeatedly refer to the importance of the hierarchy of separate grid networks in the model that vary by scale. Have they tested model performance when fewer grid networks are used? Some type of explanation of why hierarchy is important would help to maintain the conceptual thread. Is it simply that larger-scale grid networks broaden the ability to link memories/representations? 3) While neurons with “banding” patterns have been published in a prominent publication, their existence is very much in question in the field (perhaps a by-product of poor discrimination of multiple grid cells). Given this, can the authors make mention of how critical their inclusion is to model performance? Is it possible that head-direction cell patterning or head direction X border patterning (Peyrache, Nat Comm, 2017) could serve to accomplish the same thing in the model? 4) It is a bit odd that head direction cells are not included in the model. This cell type is clearly among the most impactful in driving spatial representations. Furthermore, such cells might serve a linking function for single locations with different views associated with different head orientations. This is a “link” that is not addressed in the current work.